# Reliable and Energy-Efficient LEO Satellite Communication with IR-HARQ via Power Allocation

**DOI:** 10.3390/s22083035

**Published:** 2022-04-15

**Authors:** Hongxiu Bian, Rongke Liu

**Affiliations:** 1School of Electronic and Information Engineering, Beihang University, Beijing 100191, China; bianhongxiu@buaa.edu.cn; 2Shenzhen Institution of Beihang University, Shenzhen 518038, China

**Keywords:** LEO satellite communication, energy efficiency, IR-HARQ protocol, power allocation

## Abstract

This paper examines reliable and energy-efficient transmission in low earth orbit (LEO) satellite communication systems. In particular, we analyze the link transmission characteristics of the LEO satellite to the ground user and model the channel as a combination of large-scale fading and small-scale fading. Based on this, we consider an incremental redundancy hybrid automatic repeat request (IR-HARQ) technique with a variable-power allocation method, and we call it the IR-HARQ-VPA scheme. In this method, the outage probability after each IR-HARQ round can be obtained through numerical integration based on the fast Fourier transform (NI-FFT). This method is suitable for any number of HARQ transmission rounds and can improve the accuracy compared with previous approximation methods. In addition, variable-power allocation based on the genetic algorithm (VPA-GA) is introduced to reduce the energy consumption. The simulation results show that the proposed IR-HARQ-VPA scheme cannot only meet the requirements of transmission reliability but also achieves higher energy efficiency than IR-HARQ with equal power (IR-HARQ-EP) transmission and a previously proposed variable-power allocation method. Moreover, the simulation results in a LEO satellite communication window also confirm the effectiveness of the proposed IR-HARQ-VPA scheme.

## 1. Introduction

As a supplement to terrestrial networks, sixth generation (6G) non-terrestrial networks (NTNs) can serve more users in a wide area at a low cost [1]. In 6G NTNs, low earth orbit (LEO) satellites are often used to provide services for internet of things (IoT) terminals [2,3] because of their advantages such as large coverage, low transmission delay and good environmental adaptability. In order to cope with the increasing user demand, satellite communication systems tend to operate at higher frequency bands such as Ku/Ka-band or Q/V-band, where a large number of uncongested spectrum resources are available. However, at higher frequency bands, the transmission loss of the signal will become larger and more seriously affected by the weather attenuation [4]. Rain can cause severe attenuation of more than 10 dB, resulting in signal interruption. It makes reliable transmission more challenging.

However, the energy provided by LEO satellites is severely limited [5,6]. Apart from this, in order to protect the satellite system from interference by other satellites, the maximum transmission power of the satellite is constrained [7]. Therefore, reliable transmission cannot be realized simply by increasing the transmission power. In order to improve the transmission reliability of satellite communication systems, hybrid automatic repeat request (HARQ) protocols are usually used to combat the channel fading.

Most studies on HARQ protocols in satellite communication systems mainly pay attention to the reliability or the throughput of the system. However, as the payload of a satellite is severely limited, energy efficiency should also be improved as much as possible when HARQ is adopted in the satellite communication system. In fact, many energy-efficient HARQ protocols for terrestrial communication systems have been widely studied. Inspired by this research, we focus on the study of energy-efficient HARQ in LEO satellite communication systems.

### 1.1. Related Work

Recently, there have been many studies on HARQ protocols in satellite communication systems. In [8], the throughput of the incremental redundancy HARQ (IR-HARQ) in free space optical (FSO) based satellite systems is analyzed. In [9], the author analyzes the impact of the number of parallel HARQ processes on the system memory size and throughput of a LEO satellite communication system. It also provides corresponding solutions to control the system throughput and memory size at an acceptable level. A virtual HARQ protocol is proposed in a geostationary earth orbit (GEO) satellite communication system in [10]. The virtual HARQ protocol can greatly improve the spectrum efficiency while ensuring the transmission reliability of the satellite communication system. In [11], considering fast multipath fading, type-I HARQ is adopted to achieve good bit error ratio (BER) performance in Ka-band LEO satellite mobile communication systems. In [12], the throughput performance of different HARQ protocols is analyzed in a hybrid satellite–terrestrial relay network. Moreover, the impact of HARQ transmission rounds and the information transmission rate on throughput performance is also analyzed. In [13], the author deduces the outage probability performance and diversity gain of three different HARQ protocols in the satellite–terrestrial transmission system. All the above HARQ schemes can improve the reliability of satellite communication systems. However, they do not consider the energy efficiency of HARQ in satellite communication systems which has an influence on satellite launch cost and lifetime [14].

Several energy-efficient HARQ protocols in terrestrial communication systems are proposed. An equal power transmission scheme for the HARQ protocol is designed in [15,16,17,18]. In [15], the optimal energy efficiency of the chase combining HARQ (CC-HARQ) protocol is achieved by optimizing the information length and the fixed transmission power level. In [16], the energy consumption of the CC-HARQ protocol is reduced by optimizing the maximum number of transmission rounds. In [17], under the block-fading channel, the total power used by all users in the system is minimized by optimizing the fixed transmission power of the IR-HARQ protocol. A simplified energy efficiency expression is obtained by using Gaussian approximation for the outage probability in the IR-HARQ protocol [18]. The transmission power and the IR-HARQ transmission rounds are optimized to reduce the energy consumption.

Variable-power allocation methods for IR-HARQ protocols are studied in [19,20,21,22,23,24]. In these schemes, the transmission power in each HARQ round can be adjusted. The Lagrange multiplier method is used to optimize the power sequence of the CC-HARQ protocol under quasi-static channel conditions in [19,20]. The experimental results show that the power allocation algorithm can reduce the average total power required compared to the equal power transmission scheme. In [21], the outage probability of CC-HARQ in the block-fading channel is approximated. The energy optimization is transformed into a geometric programming problem to solve. However, the approximation of outage probability is only suitable for the high signal to noise ratio (SNR) region, which causes a large error in low SNR conditions. Reference [22] proposes a suboptimal optimization algorithm to obtain the transmission power of the IR-HARQ protocol in each round. This reference only studies the special condition of two transmission rounds, which cannot be extended to the case of multiple transmission rounds. In [23,24], the author studies the energy efficiency of different HARQ protocols in time-correlated channels. By approximating the outage probability, the closed expression of each transmission power can be obtained using the Karush-Kuhn-Tucker (KKT) conditions. The simulation results show that the IR-HARQ protocol can obtain the best energy efficiency compared with other HARQ protocols. However, the approximation of the outage probability will bring a larger error compared with the exact outage probability result.

All these energy-efficient HARQ protocols are aimed at terrestrial communication systems and do not consider the complex link transmission characteristics between the LEO satellite and the ground user, especially in higher frequency bands.

### 1.2. Contributions

As the IR-HARQ has the highest diversity gain [24], we adopt it to ensure the reliable transmission of the LEO communication system in this paper. Considering the delay requirements of users, we adopt the truncated HARQ transmission strategy with a maximum of L transmission rounds. In addition, a variable-power allocation scheme is proposed to improve the energy efficiency of the LEO satellite communication system. The main contributions are summarized as follows.

The link transmission characteristics between the LEO satellite and the ground user in higher frequency bands are analyzed. We model the transmission link as a combination of large-scale fading and small-scale fading. In large-scale fading, the attenuation caused by different weather is calculated by referring to the recommendation of ITU-R [25,26,27,28]. Moreover, the carrier to noise ratio (C/N) between the LEO satellite and the ground user is simulated by the Satellite Tool Kit (STK) [29].In IR-HARQ, since the outage probability after each round cannot be expressed by an exact closed-form analytical expression, a numerical integration method based on the fast Fourier transform (NI-FFT) is proposed to calculate it quickly. The calculation results obtained by this method are more accurate than the previous method based on approximation.An IR-HARQ with a variable-power allocation (IR-HARQ-VPA) scheme is proposed. In this scheme, the variable-power allocation method based on a genetic algorithm (VPA-GA) is presented to optimize the transmission power in each IR-HARQ round. The proposed IR-HARQ-VPA scheme has higher energy efficiency than the IR-HARQ with equal power (IR-HARQ-EP) scheme and achieves reliable transmission.

In addition, numerical simulation results are presented to demonstrate the improved performance achieved by the proposed IR-HARQ-VPA scheme in a LEO satellite communication window.

### 1.3. Outline

The rest of the paper is organized as follows. In Section 2, the system model and the energy efficiency of different IR-HARQ methods are introduced. In Section 3, the NI-FFT method is proposed to calculate the outage probability of the IR-HARQ and the VPA-GA scheme is presented to maximize the energy efficiency. Simulation results are provided in Section 4. Finally, the conclusion is drawn in Section 5.

## 2. Preliminaries

### 2.1. Link Transmission Characteristics between the LEO Satellite and the Ground User

The notations in Section 2.1 are given in Table 1. As shown in Figure 1, when a LEO satellite flies into the coverage area, it will transmit information to the ground user. The fading model of the LEO satellite to the ground user includes the large-scale fading and the small-scale fading. The large-scale fading reflects the variation in the received signal amplitude over distance d and the elevation angle φ, while the small-scale fading is represented by the multipath caused by environmental scattering.

The large-scale fading between the LEO satellite and the ground user is mainly caused by two aspects. On the one hand, due to the mobility of the LEO satellite, the distance d and elevation angle φ between the satellite and the ground user are time-varying in a communication window. Therefore, the free-space path loss changes dynamically. On the other hand, when LEO satellites operate at higher frequency bands, the attenuation caused by different weather cannot be ignored.

The free-space path loss can be expressed as [30]:(1)$$L_{fs}={\left(4\pi df/c\right)}^{2},$$
where f is the carrier frequency of the system and c represents the speed of light with a value of $3.0\times 10^{8}$ m/s. The above formula can be further expressed in the form of dB:(2)$${\left(L_{fs}\right)}_{dB}=92.44+20\log d+20\log f,$$
where the unit of f is GHz and the unit of d is km.

When LEO satellites operate at higher frequency bands (above 10 GHz), the attenuation caused by atmospheric gases, clouds, fog and rain cannot be ignored, especially at low elevation angle. ITU-R provides a variety of methods to calculate or predict different weather attenuations when the elevation angle φ satisfies $5^{{}^{\circ}}\leq \varphi \leq 90^{{}^{\circ}}$ [25,26,27,28].

The attenuation caused by the atmospheric gases can be expressed as [25]:(3)$${\left(A_{g}\right)}_{dB}=\frac{{\left(A_{o}\right)}_{dB}+{\left(A_{w}\right)}_{dB}}{\sin\varphi },$$
where $A_{o}$ and $A_{w}$ reflect the atmospheric absorption caused by the oxygen and the water vapor, respectively. Both are related to the frequency f (GHz) and we call $\left(A_{o}+A_{w}\right)$ the characteristic parameter of the atmosphere absorption.

The attenuation caused by the clouds and fog is given by [26]:(4)$${\left(A_{c}\right)}_{dB}=\frac{L_{w}K_{l}^{*}(f,273.15)}{\sin\varphi },$$
where $L_{w}$ is the total columnar content of liquid water and $K_{l}^{*}$ is cloud liquid water specific attenuation coefficient. We call $L_{w}$ the characteristic parameter of the clouds and fog.

The attenuation caused by rain can be obtained using the following steps. Firstly, the specific attenuation $\gamma_{R}$ can be calculated according to [27]:(5)$$\gamma_{R}=k_{r}R_{w}{}^{\alpha_{r}},$$
where $R_{w}$ is the rainfall rate of the location, $k_{r}$ and $\alpha_{r}$ are the frequency-dependent coefficients provided in [27]. Then, the rain attenuation can be expressed as [28]:(6)$${\left(A_{r}\right)}_{dB}=\gamma_{R}L_{E},$$
where $L_{E}$ is the effective path length and related to the elevation angle φ. It can be calculated according to [28]. Similarly, we call $R_{w}$ the characteristic parameter of the rain.

The total attenuation caused by weather can be expressed as:(7)$${\left(A_{t}\right)}_{dB}={\left(A_{g}\right)}_{dB}+{\left(A_{c}\right)}_{dB}+{\left(A_{r}\right)}_{dB},$$

From the above analysis, it can be seen that the weather attenuation is related to the frequency f, elevation angle φ, and the characteristic parameter of the specific weather.

According to [31], we can write the link equation between the LEO satellite and the ground user as:(8)$${\left(C/N\right)}_{dB}={\left(EIRP\right)}_{dBW}+{\left(\frac{G_{r}}{T}\right)}_{dB/K}-{\left(k\right)}_{dBJ/K}-{\left(B_{n}\right)}_{dBHz}-{\left(L_{fs}\right)}_{dB}-{\left(L_{o}\right)}_{dB},$$
where C/N is the carrier to noise ratio, $G_{r}$ is the receive antenna gain, T is the receive noise temperature, k is Boltzman’s constant, and $B_{n}$ is the carrier bandwidth. $L_{o}$ is other transmission loss. Here, it is assumed that $L_{o}$ only includes the weather attenuation, that is, $L_{o}=A_{t}$. The term ${\left(EIRP\right)}_{dBW}={\left(P_{t}\right)}_{dBW}+{\left(G_{t}\right)}_{dBi}$ (dB), where $P_{t}$ is the transmission power and $G_{t}$ is the transmit antenna gain. Note that the values of C/N and SNR are equal when carrier suppressed modulation methods such as Binary Phase Shift Keying (BPSK) are adopted.

Based on the above analysis, we use the STK to simulate the large scale-fading between the LEO satellite and the ground user with a time interval of 5 s. The relevant parameters of LEO satellite orbit are shown in Table 2. The location of the ground user is 116°59′ E, 40°3′ N. The system operates in Ka-band with f=25 GHz. The values of ${\left(EIRP\right)}_{dBW}$ and ${\left(G/T\right)}_{dB/K}$ are 40 dBW and 20 dB/K, respectively. In addition, BPSK is adopted as the modulation mode and the C/N is equal to the value of the received SNR.

As shown in Figure 2, the LEO satellite suffers from highly variable channel conditions. While working around the horizon area, the LEO satellite usually achieves a poor C/N and the value grows larger as the satellite moves towards the ground user. We assume the characteristic parameter of the specific weather is fixed in a LEO communication window and the weather attenuation is only related to the elevation angle φ.

It is worth noting that the large-scale fading of satellites can be obtained in advance before information transmission [32]. On the one hand, the flight trajectory of the satellite is known to both the ground user and the satellite; hence, the satellite can obtain the real-time distance and the elevation angle. On the other hand, the characteristic parameters of the specific weather can be obtained from a local weather station. Therefore, the satellite can calculate the large-scale fading by (2)–(8) and estimate the value of C/N. Even without the above assumptions, we can obtain the statistics of the current channel through the feedback of the receiver. This is also used in DVB-S2 standards where a 1 s updating time is considered achievable [33].

In addition, the characteristic parameters of the specific weather change slowly. An experimental result of a satellite link reported that the average fade slope estimated from a Ku-band satellite system was 0.24 dB/second and the maximum fade slope corresponding to heavy rain events at Ka-band does not exceed 0.5 dB/second [34]. The round trip time (RTT) of the LEO to the ground is less than 20 ms [35] and the maximum number of HARQ rounds L will not be too large considering the delay requirement. Therefore, it is reasonable that we assume that the large-scale fading is a constant during the L IR-HARQ rounds but can change slowly during the whole communication window.

The small-scale fading mainly refers to the multipath fading. The amplitudes of the receiving signals follow the Rayleigh distribution which corresponds to urban areas while the Rician distribution usually matches suburban and rural areas. Since the Rayleigh distribution is a special case of the Rician distribution, we use Rician fading as the small-scale fading model in this paper for theoretical analysis which is also used in [10]. The signals received in the i th round are given as:(9)$$y_{i}=\sqrt{P_{i}}h_{l}(d_{i},\varphi_{i},f)h_{s}^{i}x_{i}+n_{i},1\leq i\leq L,$$
where $x_{i}$, $y_{i}$, and $n_{i}$ are the transmitted signal, the received signal, and the additive noise, respectively. The noise $n_{i}$ is independent and identically distributed (i.i.d) with $n_{i}\sim CN(0,\sigma_{n}^{2})$, while $P_{i}$ is the transmission power in the i th HARQ round. $h_{l}$ represents the large-scale fading while $h_{s}^{i}$ represents the small-scale fading with a Rician distribution. Based on the above analysis of the large-scale fading, the value of $h_{l}$ can be predicted for the satellite in advance and is constant during the L HARQ rounds. Let $\gamma_{i}=||h_{s}^{i}|{|}^{2}$, then the probability density function (PDF) of random variable (RV) $\gamma_{i}$ is given as:(10)$$f_{\gamma_{i}}(x)=\frac{(K+1)e^{-K}}{{\bar{\gamma }}_{i}}\cdot e^{-\frac{(K+1)x}{{\bar{\gamma }}_{i}}}\cdot I_{0}\left(2\sqrt{\frac{K(K+1)x}{{\bar{\gamma }}_{i}}}\right),x>0,$$
where K is the Rician factor defined as the ratio of the power of the LOS component to the scattered components. ${\bar{\gamma }}_{i}$ is the mean value of RV $\gamma_{i}$ and $I_{0}(\cdot )$ is the zero-order modified Bessel function of the first kind. We assume a Rician block-fading channel where the RV $h_{s}^{i}$ is a constant within one HARQ round and different between different HARQ rounds, but with the same probability distribution. Moreover, for the small-scale fading, the transmitter does not know the real value of the channel fading coefficient, only the statistical information is available.

### 2.2. Energy Efficiency in Different IR-HARQ Protocols

The application scenario shown in Figure 1 can be simplified into an end-to-end transmission block diagram as shown in Figure 3. $N_{b}$ information bits $\left(b_{1},b_{2},\ldots ,b_{N_{b}}\right)$ are encoded into $LN_{s}$ symbols. Each encoded sequence $c_{i}$ contains $N_{s}$ symbols, so the transmission rate is $R=\frac{N_{b}}{N_{s}}$ in each HARQ round. The sender transmits the coded symbol sequence $c_{i}$ with power $P_{i}$ in the i th (1≤i≤L) round. The receiver combines ($c_{1}$, $c_{2}$, …, $c_{i}$) to decode the information bits. The transmit power $P_{i}$ can either be constant or change with the HARQ round i. When the packet is decoded successfully, an acknowledgment (ACK) signal will be fed back to the sender, and the sender starts to transmit next packet. Otherwise, an negative acknowledgment (NACK) signal will be fed back and the sender will transmit the coded symbol sequence $c_{i+1}$ with the power $P_{i+1}$ until the packet is decoded or the maximum number of transmissions L is reached.

Similar to [36], we define the energy efficiency $\eta_{EE}$ as the ratio of the average number of correctly decoded information bits K to the average total consumed energy E, i.e.,
(11)$$\eta_{EE}=\frac{K}{E}\left(\mathrm{bits}/J\right).$$

Subsequently, we discuss the energy efficiency of two different IR-HARQ transmission schemes.

#### 2.2.1. IR-HARQ with Equal Power Transmission

In the IR-HARQ-EP transmission scheme, we have $P_{1}=P_{2}=\ldots =P_{L}=P$. $N_{s}$ symbols are transmitted with power P in each HARQ round. The receiver decodes the original information bits with all the received symbols. We assume the IR-HARQ protocol approaches the capacity of the channel, so the accumulated mutual information $I_{k}^{ep}$ after k rounds is:(12)$$I_{k}^{ep}=\sum_{i=1}^{k}\log_{2}\left(1+\frac{Ph_{l}{}^{2}\gamma_{i}}{\sigma_{n}^{2}}\right).$$

We define ${ℜ}_{k}≜\{I_{k}^{ep}\geq R\}$ and ${ℜ}_{k}^{c}≜\{I_{k}^{ep}<R\}$ as the decoding success event and the decoding failure event after k rounds respectively. For the IR-HARQ protocol, we have ${ℜ}_{0}\subseteq {ℜ}_{1}\subseteq \ldots \subseteq {ℜ}_{k}$ and ${ℜ}_{0}=\emptyset$. Suppose $T^{ep}$ is the total number of transmission rounds at the end of a packet transmission; then we have:(13)$$\mathrm{Pr}\{T^{ep}=k\}=\mathrm{Pr}\left\{{ℜ}_{1}^{c}\cap {ℜ}_{2}^{c}\ldots \cap {ℜ}_{k}\right\},$$
(14)$$\mathrm{Pr}\{T^{ep}=L\}=\mathrm{Pr}\left\{{ℜ}_{1}^{c}\cap {ℜ}_{2}^{c}\ldots \cap {ℜ}_{L-1}^{c}\right\}.$$

The average total energy used for the IR-HARQ-EP scheme at the end of a packet transmission is:(15)$$E^{ep}=N_{s}\sum_{i=1}^{L}\left(i\cdot P\cdot \mathrm{Pr}\left\{T^{ep}=i\right\}\right).$$

Let $e_{k}^{ep}$ denote the probability of a decoding failure after k transmission rounds; then we have:(16)$$\begin{array}{ll} e_{k}^{ep} & =\mathrm{Pr}\{{ℜ}_{1}^{c}\cap {ℜ}_{2}^{c}\ldots \cap {ℜ}_{k}^{c}\}\\ & =\mathrm{Pr}\{{ℜ}_{k}^{c}\}. \end{array}$$

According to the expression of ${ℜ}_{k}^{c}$, we have:(17)$$\begin{array}{ll} e_{k}^{ep} & =\mathrm{Pr}\{I_{k}^{ep}<R\}\\ & =\mathrm{Pr}\left\{\sum_{i=1}^{k}\log_{2}\left(1+\frac{Ph_{l}{}^{2}\gamma_{i}}{\sigma_{n}^{2}}\right)<R\right\}, \end{array}$$
by combining (13), (14), and (17), we get:(18)$$\mathrm{Pr}\{T^{ep}=k\}=e_{k}^{ep}-e_{k-1}^{ep},$$
(19)$$\mathrm{Pr}\{T^{ep}=L\}=e_{L-1}^{ep}.$$

In the IR-HARQ-EP scheme with the maximum transmission rounds L, K can be expressed as:(20)$$K^{ep}=N_{b}(1-e_{L}^{ep}).$$

The optimization problem of the IR-HARQ-EP transmission scheme can be expressed as follows:(21)$$\begin{array}{l} \underset{P}{\arg\max}\;\eta_{EE}^{ep}=\frac{R\left(1-e_{L}^{ep}\right)}{\sum_{i=1}^{L-1}iP\left(e_{i-1}^{ep}-e_{i}^{ep}\right)+e_{L-1}^{ep}LP}\\ s.t.\;\left\{\begin{array}{c} e_{L}^{ep}⩽\epsilon_{0}\\ 0<P\leq P_{\max},1\leq i\leq L, \end{array}\right. \end{array}$$
where $\epsilon_{0}$ is the maximum outage probability allowed by the communication system and $P_{max}$ is the maximum transmission power of the transmitter. We use $\eta_{EE}^{ep}$ to measure the energy efficiency of the IR-HARQ-EP transmission scheme.

#### 2.2.2. IR-HARQ with Variable Power Transmission

Different from the above fixed power transmission scheme, we adjust the transmission power $P_{i}$ ($P_{i}\neq P_{j},i\neq j$) in each HARQ round to improve the energy efficiency under the constraint of a desired outage probability $\epsilon_{0}$. In the IR-HARQ-VPA transmission scheme, the mutual information $I_{k}^{vpa}$ accumulated after k rounds is:(22)$$I_{k}^{vpa}=\sum_{i=1}^{k}\log_{2}\left(1+\frac{P_{i}h_{l}{}^{2}\gamma_{i}}{\sigma_{n}^{2}}\right).$$

The average total energy used for the IR-HARQ-VPA transmission scheme is:(23)$$E^{vpa}=N_{s}\sum_{i=1}^{L}\left(\sum_{j=1}^{i}P_{j}\right)\mathrm{Pr}\left\{T^{vpa}=i\right\}.$$

The probability of decoding failure after k transmission rounds $e_{k}^{vpa}$ is:(24)$$\begin{array}{ll} e_{k}^{vpa} & =\mathrm{Pr}\left\{I_{k}^{vpa}<R\right\}\\ & =\mathrm{Pr}\left\{\sum_{i=1}^{k}\log_{2}\left(1+\frac{P_{i}h_{l}{}^{2}\gamma_{i}}{\sigma_{n}^{2}}\right)<R\right\}. \end{array}$$

Similar to (20), we have $K^{vpa}=N_{b}(1-e_{L}^{vpa})$. The optimization problem of the IR-HARQ-VPA transmission scheme can be given as follows:(25)$$\begin{array}{l} \underset{\left(P_{1},\ldots ,P_{L}\right)}{\arg\max}\;\eta_{EE}^{vpa}=\frac{R\left(1-e_{L}^{vpa}\right)}{\sum_{i=1}^{L-1}\left(\sum_{j=1}^{i}P_{j}\right)\left(e_{i-1}^{vpa}-e_{i}^{vpa}\right)+e_{L-1}^{vpa}\sum_{i=1}^{L}P_{i}}\\ s.t.\;\left\{\begin{array}{c} e_{L}^{vpa}⩽\epsilon_{0}\\ 0<P_{i}\leq P_{\max},1\leq i\leq L. \end{array}\right. \end{array}$$

## 3. Outage Analysis and Solution to the Optimization Problems

To maximize the energy efficiency, the key is to calculate the outage probability in (17) and (24). In this section, we propose the NI-FFT method to calculate it accurately. Based on this, the VPA-GA method is designed to optimize the transmission power in each IR-HARQ round.

### 3.1. Outage Probability of IR-HARQ

To simplify the analysis, let ${\bar{\gamma }}_{i}=1$ and $\sigma_{n}^{2}=1$; then, $P_{i}h_{l}{}^{2}$ and $P_{i}h_{l}{}^{2}\gamma_{i}$ represent the average SNR and the instantaneous SNR received in the i th HARQ round, respectively. The outage probability of the IR-HARQ protocol after k rounds can be calculated as follows:(26)$$e_{k}^{IR}=\mathrm{Pr}\left\{\frac{1}{R}\sum_{i=1}^{k}\log_{2}\left(1+P_{i}h_{l}{}^{2}\gamma_{i}\right)<1\right\}.$$

Let $X_{i}=\frac{1}{R}\log_{2}(1+P_{i}h_{l}{}^{2}\gamma_{i})$; then, $X_{1},X_{2},\ldots ,X_{L}$ are i.i.d. The probability density function (PDF) of RV $X_{i}$ is as follows:(27)$$f_{i}(x_{i})=\mu 2^{Rx_{i}}\cdot e^{-(K+1)\frac{2^{Rx_{i}}-1}{P_{i}h_{l}{}^{2}}}\cdot I_{0}\left(\sqrt{\nu \frac{2^{Rx_{i}}-1}{P_{i}h_{l}{}^{2}}}\right),x_{i}>0,$$
where $\mu =\frac{(K+1)R\ln2}{P_{i}h_{l}{}^{2}}e^{-K}$ and ν=4K(K+1). Therefore, the outage probability can be calculated by the following integral expression:(28)$$e_{k}^{IR}=\int \ldots \int_{D_{k}}f_{1}(x_{1})f_{2}(x_{2})\ldots f_{k}(x_{k})dx_{k}\ldots dx_{2}dx_{1},$$
where $D_{k}:X_{1}+X_{2}+\ldots +X_{k}<1$. (28) can be further expressed as:(29)$$e_{k}^{IR}=\int_{0}^{1}\ldots \int_{0}^{1-\sum_{j=1}^{k-1}x_{j}}\prod_{i=1}^{k}f_{i}(x_{i})dx_{k}\ldots dx_{2}dx_{1},$$

As the integration in (29) is not an elementary function, the right side of (29) cannot be expressed by an exact closed-form analytical expression. To calculate (29), the multiple numerical integration (MNI) method is usually used. However, it is only suitable for L with small values. When L increases, the computational complexity increases exponentially. Moreover, in (29), the upper limit of some integral variables is not a constant, but a function of other integral variables.

In order to solve this problem, the outage probability of the IR-HARQ after k rounds in different fading channels are approximated in [13,24]. However, they have a large approximation error, which will mean that either the system cannot meet the reliability requirements or the energy efficiency of the system is decreased. The large approximation error will mean that either the system cannot meet the reliability requirements or the energy efficiency of the system is decreased. Therefore, a new and effective method is needed to calculate the outage probability effectively.

Rethinking (26), let $Y_{k}=\sum_{i=1}^{i=k}X_{i}$. Suppose $g_{k}(y)$ as the pdf of the new variable $Y_{k}$; then, $g_{k}(y)$ can be calculated as:(30)$$g_{k}(y)=f_{1}(x_{1})⊗f_{2}(x_{2})⊗\ldots ⊗f_{k}(x_{k}),$$
where ⊗ denotes the convolution. Then, the outage probability $e_{k}^{IR}$ can be calculated by the following:(31)$$e_{k}^{IR}=\int_{0}^{1}g_{k}(y)dy.$$

Equation (30) can be calculated quickly by FFT [37]. Firstly, we take the discrete value of $f_{i}(x_{i})$ on (0,1] according to the discrete interval Δ, i.e.,
(32)$$f_{i}(x_{n})=\mu 2^{Rx_{n}}\cdot e^{-(K+1)\frac{2^{Rx_{n}}-1}{P_{i}h_{l}{}^{2}}}\cdot I_{0}\left(\sqrt{\nu \frac{2^{Rx_{n}}-1}{P_{i}h_{l}{}^{2}}}\right),$$
where $x_{n}=\delta ,\delta +\Delta ,\delta +2\Delta ,\ldots 1,$ δ takes a small value since the domain of variable $x_{i}$ is greater than 0. Suppose an N-point FFT is implemented; then, we have:(33)$$g_{k}(x_{n})=\mathrm{IFFT}\left(\prod_{i=1}^{k}\mathrm{FFT}\left(f_{i}(x_{n}),N\right),N\right)\times \Delta^{k-1},$$
where the IFFT is the inverse operation of FFT. Finally, we calculate the value of outage probability $e_{k}^{IR}$ by numerical integration according to (31).

### 3.2. Proposed Energy Efficiency Maximization

In this section, we maximize the energy efficiency of the IR-HARQ-EP and the IR-HARQ-VPA schemes. The key is to optimize the power P or the power sequence $\left(P_{1},P_{2},\ldots ,P_{L}\right)$ to obtain the maximum energy efficiency $\eta_{EE}^{ep}$ and $\eta_{EE}^{vpa}$ in (21) and (25), respectively. The two optimization problems can be simplified as follows:

**Problem** **1.***In the IR-HARQ-EP scheme, the energy efficiency*$\eta_{EE}^{ep}$*is maximized over transmission power*P, *which is: *(34)$$\begin{array}{l} \underset{P}{\arg\max}\;\eta_{EE}^{ep}=\frac{R\left(1-e_{L}^{ep}\right)}{\sum_{i=1}^{L}e_{i-1}^{ep}P}\\ s.t.\;\left\{\begin{array}{c} e_{L}^{ep}⩽\epsilon_{0}\\ 0<P\leq P_{\max},1\leq i\leq L. \end{array}\right. \end{array}$$

**Problem** **2.***In the IR-HARQ-VPA scheme, the energy efficiency*$\eta_{EE}^{vpa}$*is maximized over the transmission power sequence*$\left(P_{1},P_{2},\ldots ,P_{L}\right)$, *which is: *(35)$$\begin{array}{l} \underset{\left(P_{1},P_{2},\ldots ,P_{L}\right)}{\arg\max}\;\eta_{EE}^{vpa}=\frac{R\left(1-e_{L}^{vpa}\right)}{\sum_{i=1}^{L}e_{i-1}^{vpa}P_{i}}\\ s.t.\;\left\{\begin{array}{c} e_{L}^{vpa}⩽\epsilon_{0}\\ 0<P_{i}\leq P_{\max},1\leq i\leq L, \end{array}\right. \end{array}$$*where*$\epsilon_{0}$*is the maximum outage probability allowed by the communication system and*$P_{max}$*is the maximum transmission power. At HARQ round 0, the values of*$e_{0}^{ep}$*and*$e_{0}^{vpa}$*are set to 1 by default.*

As the exact closed-form analytical expression of $e_{i}$ cannot be obtained, the optimal energy efficiency $\eta_{EE}^{vpa}$ is intractable mathematically. Therefore, we propose a heuristic method based on the genetic algorithm (GA) to solve the above problems.
**Algorithm 1** Variable power allocation based on the GA1:Initialization: transmission rate R$;\;\mathrm{outage}\;\mathrm{probability}\;\epsilon_{0}$; maximum transmission times L$;\;\mathrm{power}\;\mathrm{range}\;P_{max}$; iterations N$;\;\mathrm{crossover}\;\mathrm{probability}\;p_{c}$$;\;\mathrm{variation}\;\mathrm{probability}\;p_{v}$$;\;\mathrm{population}\;\mathrm{size}\;size_{pop}$;2:Initial population pop$\;\mathrm{in}\;(0,P_{max})$;3:for i=1 to N do4:  
$\mathrm{calculate}\;e_{1},e_{2},\ldots ,e_{L}$
 of all individuals from pop using NI-FFT method;5:  
$\mathrm{calculate}\;\eta_{EE}^{vpa}$ of all individuals from pop using equation (34);6:  
for j=1:2 to N do7:    
$pop{\_}_{parent}\leftarrow$ select two parents from pop$\;\mathrm{according}\;\mathrm{to}\;\eta_{EE}^{vpa}$;8:    
$pop{\_}_{bin}\leftarrow$$\;\mathrm{coding}\;\mathrm{result}\;\mathrm{of}\;pop{\_}_{parent}$;9:    
$pop{\_}_{cross}\leftarrow$$\;\mathrm{crossing}\;\mathrm{result}\;\mathrm{of}\;pop{\_}_{bin}$;10:    
$pop{\_}_{var}\leftarrow$$\;\mathrm{mutating}\;\mathrm{result}\;\mathrm{of}\;pop{\_}_{cross}$;11:    
$pop{\_}_{new}$(2j−1:2j,:)←$pop{\_}_{var}$;12:  end for13:  
$pop{\_}_{dec}\leftarrow$$\;\mathrm{decoding}\;\mathrm{result}\;\mathrm{of}\;pop{\_}_{new}$;14:  Complete an evolution and update the population15:  
pop←
$pop{\_}_{dec}$;16:end for17:$\mathrm{obtain}\;\mathrm{the}\;\mathrm{maximum}\;\mathrm{value}\;\mathrm{of}\;\eta_{EE}^{vpa*}$;18:$\mathrm{return}\;\mathrm{the}\;\mathrm{optimal}\;\mathrm{power}\;\mathrm{sequence}\;(P_{1}^{*},P_{2}^{*},\ldots ,P_{L}^{*})$;


The GA is a method to search for suboptimal solutions by simulating the natural evolution process. In the proposed VPA-GA method, a population pop with population size of $size_{pop}$ is firstly initialized, which is also an initial solution set of the optimization problem. GA evaluates each chromosome which represents an individual of pop. The chromosomes $pop_{\_parent}$ are selected based on the fitness value. In this paper, the fitness function is the same as (34) and (35). The chromosomes with high fitness have more reproductive opportunities. The selected individuals or chromosomes need to be coded; then, crossover and mutation operations are carried out in turn to get the new individuals $pop_{\_new}$ after evolution. Then, the new population begins the next round of evolution until the maximum number of evolutions is reached. The specific calculation process is described in Algorithm 1.

In the proposed VPA-GA method, we assume the maximum transmission rounds of the IR-HARQ is L, M-point FFT and IFFT are used in the calculation of $e_{i}^{IR}(i>1)$; the population size is Q and the iterations is N. Based on this, we analyze the computational complexity of the proposed VPA-GA method as shown in Algorithm 1. In each iteration, the calculation cost mainly includes the following two parts:(a)Calculate the fitness of Q individuals in the population. In the process of calculating the fitness of each individual, it is necessary to calculate the outage probability $e_{1},e_{2},\ldots ,e_{L}$. When calculating $e_{i}\;(i>1)$, M-point FFT and IFFT are used. Therefore, the time complexity of the process is $O\left(Q\cdot \left(L-1\right)\cdot 2\cdot M\cdot \log2(M)\right)$.

(b)In each evolution, two parents are selected from Q individuals for crossover, mutation, and other operations until Q individuals are updated. The computational complexity of crossover and mutation is much lower than that of M-point FFT and IFFT in (a). Therefore, compared with (a), the complexity of (b) can be ignored.

In short, the computational complexity of VPA-GA method with U iterations is about $O\left(U\cdot Q\cdot L\cdot M\cdot \log2(M)\right)$.

## 4. Algorithm Implementation and Simulation Results

In this section, numerical simulation results are presented based on the above theoretical analysis. The simulation experiment mainly includes the calculation results of outage probability using different methods and the energy efficiency of the proposed variable power allocation method.

### 4.1. Simulation Results of the Outage Probability of IR-HARQ

In this section, we calculate the outage probability of IR-HARQ using different methods. Without losing generality, we take $P_{1}=\ldots =P_{L}$ as an example. Figure 4 shows a comparison of the outage probability obtained by the approximation method in [24] and the exact value obtained by the MNI method under the Rician factor of K=0 (Rayleigh fading). When L=2 and P=1, the approximation error is more than 50% in Figure 4a. When L=3 and P=45, the approximation error is about 17% in Figure 4b.

As can be seen from Figure 4, the outage probability calculated by the NI-FFT method is almost the same as the exact value obtained by the MNI method. The results of the proposed NI-FFT method under different rates R are shown in Figure 5. It also proved that the proposed NI-FFT method can calculate the outage probability of IR-HARQ more accurately. Furthermore, when L increases, the NI-FFT method is still applicable.

### 4.2. Simulation Results of the Energy Efficiency of Different Transmission Schemes

In this section, we simulate the energy efficiency of different transmission schemes. The relevant simulation parameters are set as: $h_{l}=1$, ${\bar{\gamma }}_{i}=1$ and $\sigma_{n}^{2}=1$.

The energy efficiency of different IR-HARQ transmission schemes with various values of $\epsilon_{0}$ is shown in Figure 6 with K=5 and R=2. As shown in Figure 6, the energy efficiency of the proposed IR-HARQ-VPA scheme is much higher than that of the IR-HARQ-EP scheme. The higher the reliability that is required, the greater the energy efficiency improvement provided by the IRHARQ-VPA scheme. When L=3, $\epsilon_{0}=10^{-6}$, the energy efficiency of IR-HARQ-VPA scheme is 63.6% higher than that of the IR-HARQ-EP scheme. When the outage probability $\epsilon_{0}=0.1$, the energy efficiency of two schemes is almost the same. This is mainly because the transmission power required by the two schemes is small under this condition and the advantage of power allocation will become smaller.

Figure 7 shows the energy efficiency of two schemes under different transmission rates with K=2 and $\epsilon =10^{-3}$. With the increase in the rate R, the energy efficiency of both schemes will decrease. The energy efficiency of the IR-HARQ-VPA scheme is 44.4% higher than that of the IR-HARQ-EP scheme when R=1 and L=2. When L=4, the optimized power sequence $\left(P_{1},P_{2},P_{3},P_{4}\right)$ of the IR-HARQ-VPA scheme is shown in Figure 8 in detail. As it can be seen, the power sequence is not monotonic. The power value of the last IR-HARQ round is much greater than that of each previous round. This may be due to using small power in previous rounds to improve energy efficiency and using large power in the last round to ensure the transmission reliability under the condition that the previous decoding fails.

The energy efficiency of both schemes increases with the increase in the transmission rounds L as shown in Figure 9. This is mainly due to the increase in diversity gain caused by the increase in transmission rounds. As analyzed in Section 3.2, the complexity of the proposed scheme is $O\left(U\cdot Q\cdot L\cdot M\cdot \log2(M)\right)$, which increases with the increase in L. The energy efficiency increases with the increase in L, as shown in Figure 9. In addition, the value of L will also affect the transmission delay; when the value of L is too large, the transmission delay will increase. Therefore, the value of L should be properly selected to achieve a trade-off between energy efficiency, complexity, and transmission delay.

When $\epsilon_{0}=10^{-5}$ and L=3, the energy efficiency of the IR-HARQ-VPA scheme is 2.6 times than that of the IR-HARQ-EP scheme. In Figure 10, the energy efficiency of the proposed IR-HARQ-VPA scheme is higher than that of the IR-HARQ-EP scheme under different values of K. Especially when the channel condition is very poor with K=1, the energy efficiency of the IR-HARQ-VPA scheme is 37.1% higher than that of the IR-HARQ-EP scheme.

In [24], a variable-power allocation for the IR-HARQ scheme was studied for the Rayleigh fading case (K=0). The authors used the approximation method to calculate the outage probability, and then used KKT conditions to solve the optimization problems. We call this method VPA-Approximation. Figure 11 shows that our proposed VPA-GA method has the higher energy efficiency. This is mainly because the VPA-Approximation method has a large approximation error and affects the optimal power allocation. The energy efficiency of the VPA-GA method is 15.4%, 26.6% higher than that of the VPA-Approximation method when $\epsilon_{0}=10^{-3}$, $\epsilon_{0}=10^{-2}$. Figure 12 shows the optimization process of the VPA-GA method. The energy efficiency quickly converges to the optimal value, which also verifies the effectiveness of the proposed algorithm.

In addition, we apply the proposed IR-HARQ-VPA scheme to the LEO satellite communication system. For rainy weather, the impact on the C/N is shown in Figure 2. We assume the satellite can predict the large-scale fading information based on the satellite trajectory and the weather forecast. The Rician fading is used to model the small-scale fading. To adjust the power of different IR-HARQ rounds is equivalent to adjusting the received C/N in each round. Therefore, the value of C/N in Figure 2 is the maximum value of $P_{i}$ in (9) under the condition that $h_{l}=1$, ${\bar{\gamma }}_{i}=1$ and $\sigma_{n}^{2}=1$. In order to shorten the simulation time, we exploit the symmetry of the satellite orbit and only simulate half the time of the LEO satellite communication window.

Figure 13 shows the reliability of different transmission schemes in half of the communication window. When $\epsilon =10^{-5}$, the scheme without HARQ cannot satisfy the requirements of reliability during the whole communication window. As the channel condition is poor at low elevation angles, both the IR-HARQ-EP and the IR-HARQ-VPA schemes cannot satisfy the requirements of reliability in the early stages of the LEO satellite entering the communication window. It can also be seen from Figure 13 that when the energy efficiency reaches the maximum, the outage probability satisfies $e_{L}=\epsilon_{0}$.

Figure 14 shows the energy efficiency of different transmission schemes in half of the communication window. The maximum transmission power is assumed to be $P_{max}=300$ W. The scheme without HARQ has the smallest energy efficiency as the scheme has no diversity gain. The energy efficiency of all schemes will become higher as the channel state becomes better, especially the IR-HARQ-EP and the IR-HARQ-VPA schemes. The proposed IR-HARQ-VPA scheme achieves more than twice the energy efficiency of the IR-HARQ-EP scheme during most of the communication window.

## 5. Conclusions

In order to ensure reliable and energy-efficient transmission in the LEO satellite communication systems, we introduced an IR-HARQ-VPA scheme. In this scheme, the NI-FFT method was proposed to calculate the outage probability accurately. Based on this, the genetic algorithm was used to optimize the transmission power sequence. The energy efficiency of the proposed scheme is much higher than that of the equal power transmission scheme and the previously proposed variable-power allocation scheme. Simulation results for a realistic satellite orbit also indicate that the energy efficiency can be significantly improved by the proposed IR-HARQ-VPA scheme in a LEO satellite communication window. Our future work will take variable-rate transmission into account. Through the joint rate and power optimization in each IR-HARQ round, the energy efficiency of LEO satellite communication systems can be further improved.

## Figures and Tables

**Figure 1 sensors-22-03035-f001:**
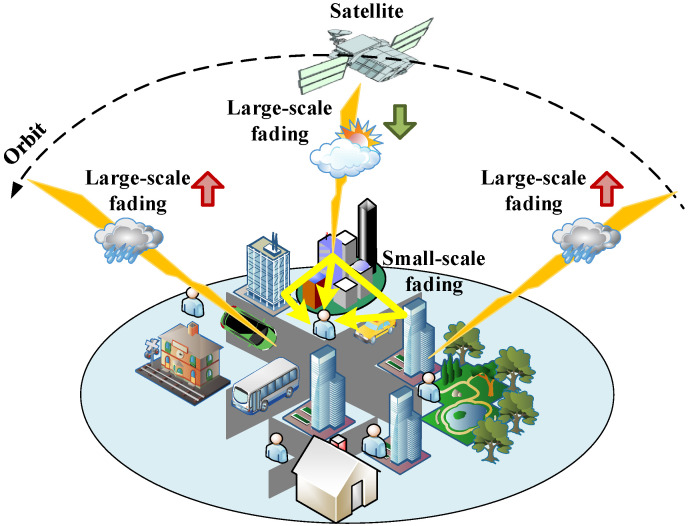
The information is transmitted from the LEO satellite to the ground user.

**Figure 2 sensors-22-03035-f002:**
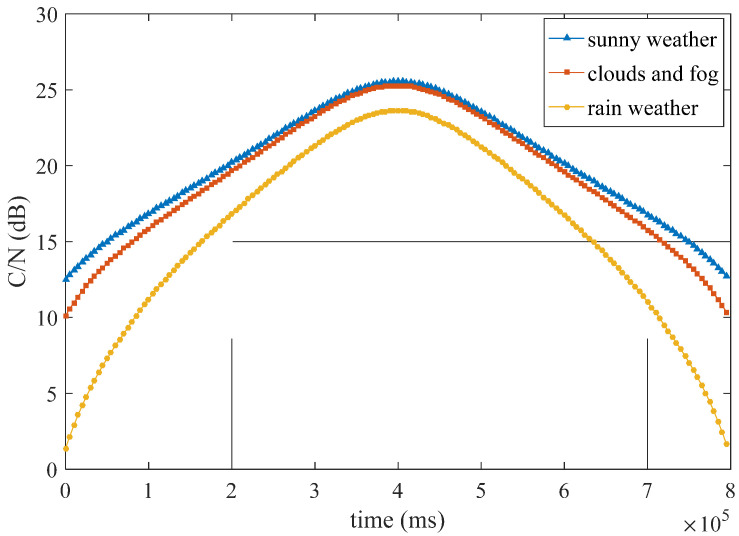
The link equation under different weather scenarios in a LEO communication window.

**Figure 3 sensors-22-03035-f003:**
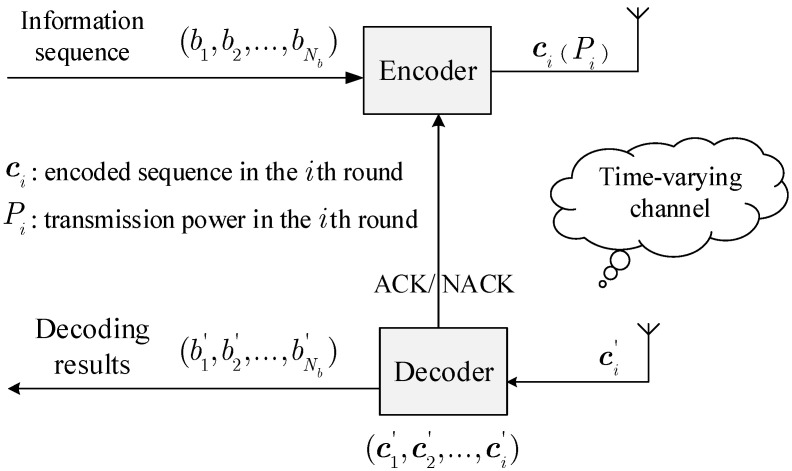
The transmission model of IR-HARQ.

**Figure 4 sensors-22-03035-f004:**
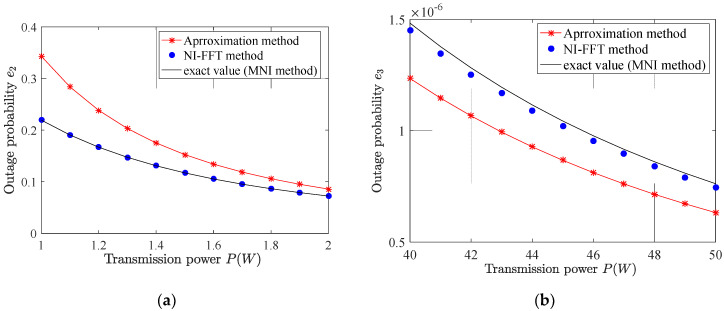
Comparison between the approximate and exact outage probability for the Rayleigh fading channel: (**a**) with K=0, R=1, L=2; (**b**) with K=0, R=1, L=3.

**Figure 5 sensors-22-03035-f005:**
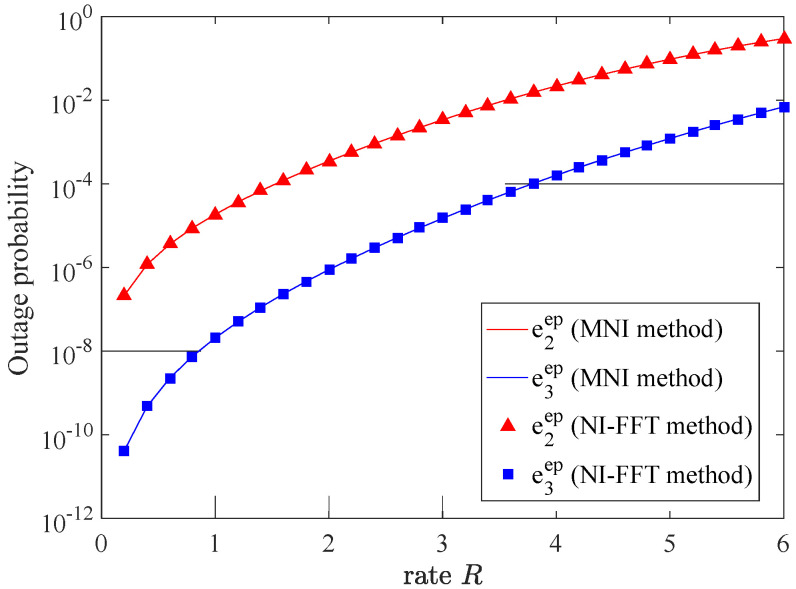
The comparison of the outage probability between the NI-FFT method and the MNI method (P=10,K=5).

**Figure 6 sensors-22-03035-f006:**
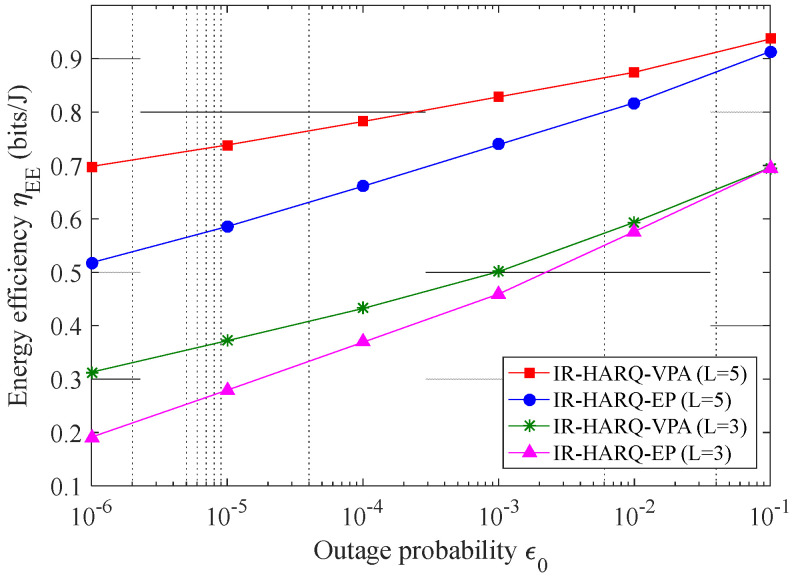
The energy efficiency of IR-HARQ-EP and IR-HARQ-VPA schemes with K=5 and R=2 under different values of $\epsilon_{0}$.

**Figure 7 sensors-22-03035-f007:**
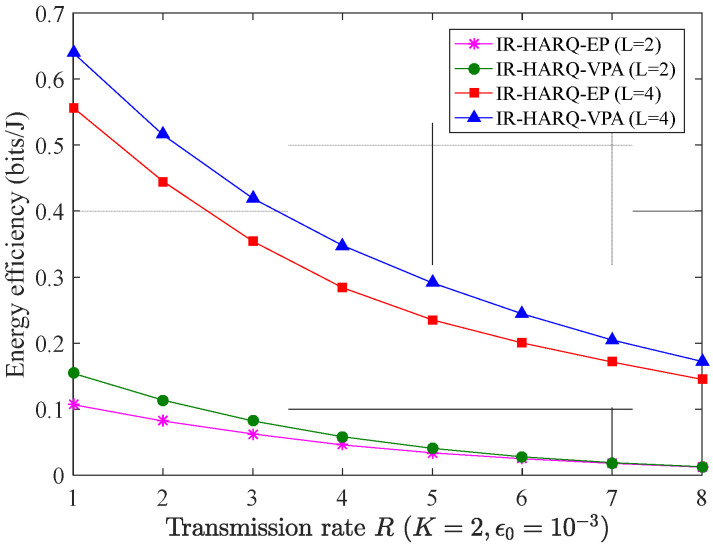
The energy efficiency of IR-HARQ-EP and IR-HARQ-VPA schemes with K=2 and $\epsilon =10^{-3}$ under different values of R.

**Figure 8 sensors-22-03035-f008:**
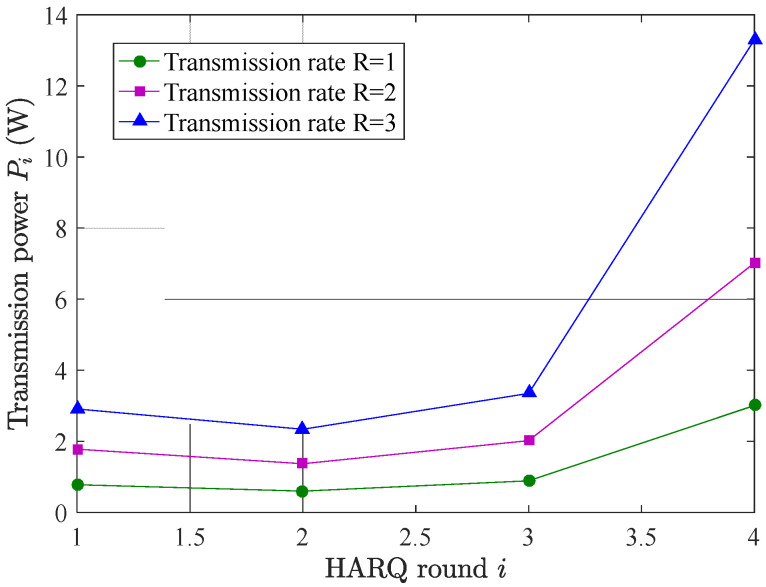
The value of $P_{i}$ (1≤i≤L) to be used in different IR-HARQ rounds for the IR-HARQ-VPA scheme $\left(K=2,\epsilon_{0}=10^{-3},L=4\right)$.

**Figure 9 sensors-22-03035-f009:**
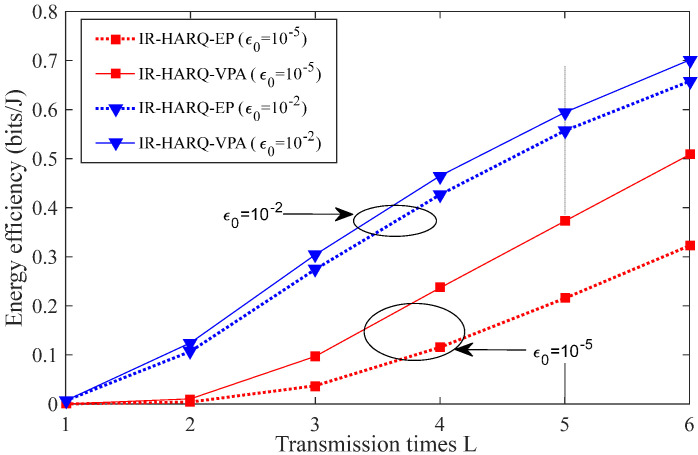
The energy efficiency of IR-HARQ-EP and IR-HARQ-VPA schemes under different values of L(R=2,K=0).

**Figure 10 sensors-22-03035-f010:**
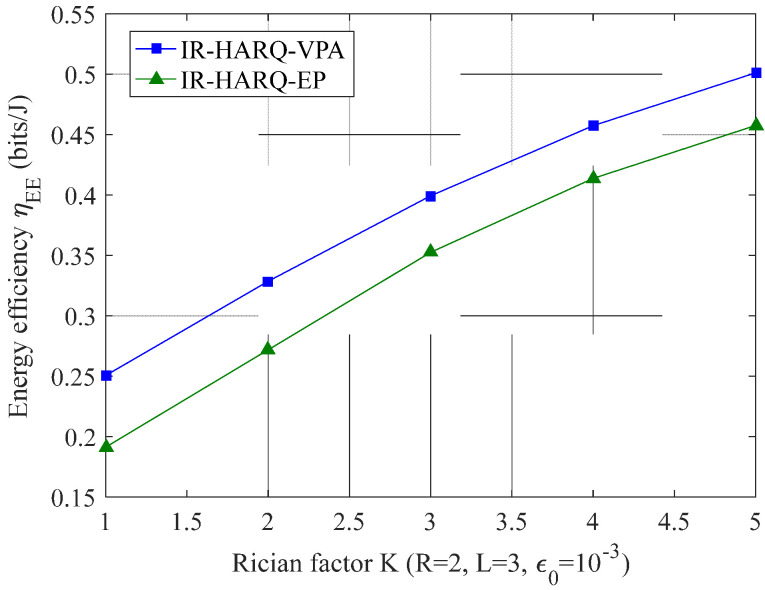
The energy efficiency of IR-HARQ-EP and IR-HARQ-VPA schemes under different Rician factors K.

**Figure 11 sensors-22-03035-f011:**
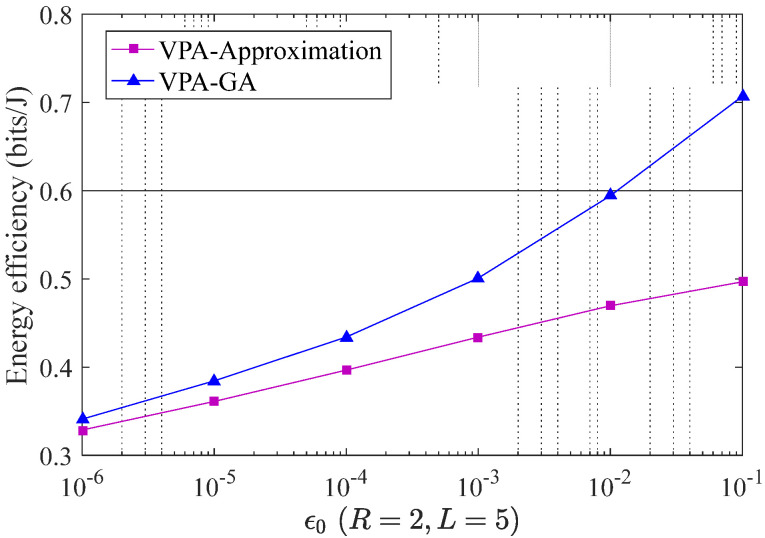
The comparison between the VPA-GA method and VPA-Asymptotic method under K=0.

**Figure 12 sensors-22-03035-f012:**
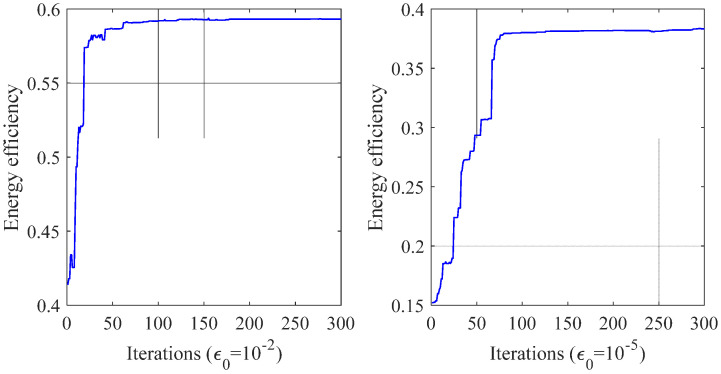
Energy efficiency optimization process based on a genetic algorithm.

**Figure 13 sensors-22-03035-f013:**
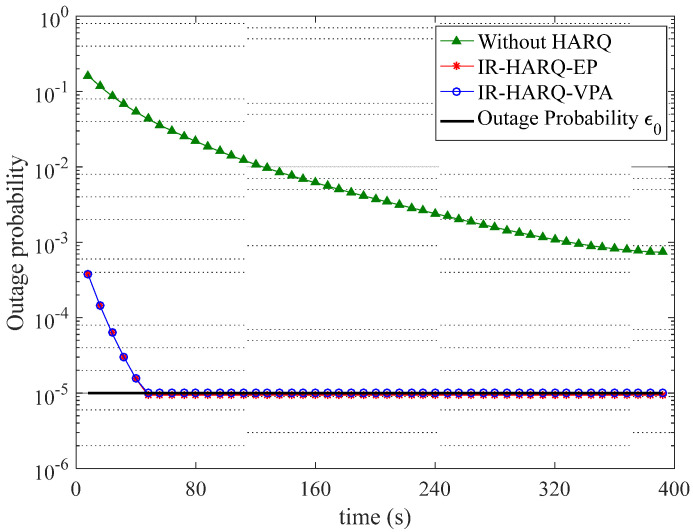
The outage probability of different transmission schemes for half of the communication window.

**Figure 14 sensors-22-03035-f014:**
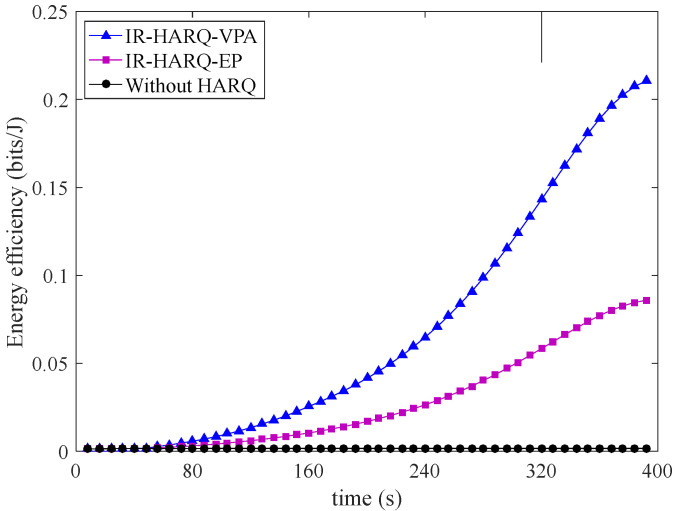
The energy efficiency of different transmission schemes in half of the communication window.

**Table 1 sensors-22-03035-t001:** Description of notations.

Notations	Description
d	Distance between the satellite and the ground user
φ	Elevation angle between the satellite and the ground user
f	Carrier frequency of the system
c	Speed of light
$L_{fs}$	Free-space path loss
$A_{o}$	Atmospheric absorption caused by the oxygen
$A_{w}$	Atmospheric absorption caused by the water vapor
$A_{g}$	Attenuation caused by the atmospheric gases
$A_{c}$	Attenuation caused by the clouds and fog
$L_{w}$	Total columnar content of liquid water
$K_{l}^{*}$	Cloud liquid water specific attenuation coefficient
$\gamma_{R}$	Specific attenuation defined in [27]
$R_{w}$	Rainfall rate
$k_{r}$	Frequency-dependent coefficient defined in [27]
$\alpha_{r}$	Frequency-dependent coefficient defined in [27]
$L_{E}$	Effective path length
C/N	Carrier to noise ratio
$G_{r}$	Receive antenna gain
$G_{t}$	Transmit antenna gain
T	Receive noise temperature
k	Boltzman’s constant
$B_{n}$	Carrier bandwidth

**Table 2 sensors-22-03035-t002:** Relevant parameters of the LEO satellite orbit.

Attribute Name	Attribute Value
Semi-major axis	7178.14 km
Eccentricity	$1.11022\;e{}^{-16}$
Inclination	45°
Argument of perigee	0°
Right Ascension of Ascending Node (RANN)	0°
True anomaly	$1.40562\;e{}^{-16}$

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
