# Peer review of "Reliable and Energy-Efficient LEO Satellite Communication with IR-HARQ via Power Allocation"

_sensors, 2022, doi:10.3390/s22083035_

Round 1

Reviewer 1 Report

In this paper,  the authors propose a redundancy hybrid automatic repeat request (IR-HARQ) technique with a optimized power allocation method  to maximize the energy efficiency. The transmission link between LEO satellites and users is first modelled and then proposed IR-HARQ with variable-power allocation scheme is shown to have higher energy efficiency in validation sections via extensive simulations. In general, the paper's technical content is suitable for publications. Moreover, the paper's presentation style is acceptable. Some organizational improvements are required.

Some comments:

- A notation table would help
- There can be some organizational changes, e.g. some of the comparison results mentioned such as Fig. 4. Fig. 5 can be put into Section 4
- How did you obtained Fig. 2 numbers?
- Fig.1 can be improved further for extending the motivation as it current version is rather simple and does not describe well the motivation of the paper.
- Fig.4 (b) has some text under caption not seen.
- What is the complexity of Algorithm 1?
- What are the trade-offs of using the proposed scheme? A discussion is needed inside the text.

Reviewer 2 Report

This paper reports that applying variable power allocation technology to IR-HARQ can improve the reliability and energy efficiency of LEO satellite communications. This is a very interesting proposal.

The distinction between dB and true values is not clearly shown for each parameter used in Chapter 2, which describes link analysis.

For example, the right-hand side of equation (7) should be written as (A_g) _dB + (A_c) _dB + (A_r) _dB. The right side of equation (8) cannot be expressed by addition unless each parameter has a dB value. Review the description of each parameter in this chapter.
